Resource

# Multiple sclerosis risk variants regulate gene expression in innate and adaptive immune cells

Melissa M Gresle[1,2,3,*], Margaret A Jordan[4,*], Jim Stankovich[3,*], Tim Spelman[1,13], Laura J Johnson[5], Louise Laverick[1], Alison Hamlett[5], Letitia D Smith[4], Vilija G Jokubaitis[1,3], Josephine Baker[6], Jodi Haartsen[7], Bruce Taylor[8], Jac Charlesworth[8], Melanie Bahlo[9,10], Terence P Speed[11], Matthew A Brown[12,14], Judith Field[5,*], Alan G Baxter[4,*], Helmut Butzkueven[3,*]

**At least 200 single-nucleotide polymorphisms (SNPs) are associated with multiple sclerosis (MS) risk. A key function that could mediate SNP-encoded MS risk is their regulatory effects on gene expression. We performed microarrays using RNA extracted from purified immune cell types from 73 untreated MS cases and 97 healthy controls and then performed Cis expression quantitative trait loci mapping studies using additive linear models. We describe MS risk expression quantitative trait loci associations for 129 distinct genes. By extending these models to include an interaction term between genotype and phenotype, we identify MS risk SNPs with opposing effects on gene expression in cases compared with controls, namely, rs2256814 _MYT1_ in CD4 cells (q = 0.05) and rs12087340 _RF00136_ in monocyte cells (q = 0.04). The rs703842 SNP was also associated with a differential effect size on the expression of the _METTL21B_ gene in CD8 cells of MS cases relative to controls (q = 0.03). Our study provides a detailed map of MS risk loci that function by regulating gene expression in cell types relevant to MS.**

## Introduction

Multiple Sclerosis (MS) is an autoimmune disease causing multifocal central nervous system inflammatory demyelination and axonal injury. The etiology of MS is unknown, but current evidence suggests that complex interactions between environmental risk factors and common genetic variants determine MS susceptibility (reviewed in Olsson et al [2017]).

Support for a genetic contribution to MS is largely derived from epidemiological studies that show an association between ancestry and MS prevalence, and evidence for familial clustering (Ebers et al, 1986; Sadovnick & Baird, 1988; Pugliatti et al, 2001; Baranzini & Oksenberg, 2017). Linkage studies of families with multiple affected members with MS show that variation within the human leukocyte antigen (HLA) class II locus on chromosome 6p21 is associated with an increased risk of MS, with the main risk allele, _HLA-DRB1*15:01_, carrying an average odds ratio of 3.08 (Bertrams & Kuwert, 1976; Hollenbach & Oksenberg, 2015). More recent genome-wide association studies have identified at least 200 single nucleotide polymorphisms (SNPs) outside of the MHC region that are associated with MS risk (International Multiple Sclerosis Genetics Consortium et al 2011, 2013, 2019; ). These SNPs are all common (>2.1% risk allele frequency), are also present in unaffected individuals, and have small odds ratios of between 0.8 and 1.3, representing both protective and risk variants. Hence, carriage of these common genetic variants is unlikely to be sufficient or necessary for the pathogenesis of disease.

Importantly, these non-HLA MS risk variants are mainly found in intronic or intragenic regions, and so their functional effects are largely unknown (International Multiple Sclerosis Genetics Consortium et al, 2013). It is recognized, however, that common noncoding variants associated with traits and diseases, including MS, are enriched within DNAse I hypersensitivity sites and other marks that signify the presence of regulatory DNA, including promoter and enhancer regions (Maurano et al, 2012; Elangovan et al, 2014). Genetic loci associated with differential gene expression are referred to as expression quantitative trait loci (eQTL), with those acting locally termed cis-eQTL and those

[1]Department of Medicine, University of Melbourne, Parkville, Australia   [2]Melbourne Brain Centre, Royal Melbourne Hospital, University of Melbourne, Parkville, Australia   [3]Department of Neuroscience, Central Clinical School, Monash University, Melbourne, Australia   [4]Molecular & Cell Biology, James Cook University, Townsville, Australia   [5]Florey Institutes of Neuroscience and Mental Health, Parkville, Australia   [6]Multiple Sclerosis Clinical & Research Unit, Melbourne Health, Royal Melbourne Hospital, Parkville, Australia   [7]Eastern Clinical Research Unit, Eastern Health, Box Hill, Australia   [8]Menzies Institute for Medical Research, University of Tasmania, Hobart, Australia   [9]Population Health and Immunity Division, The Walter and Eliza Hall Institute of Medical Research, Parkville, Australia   [10]Department of Medical Biology, The University of Melbourne, Parkville, Australia   [11]Bioinformatics Division, The Walter and Eliza Hall Institute of Medical Research, Parkville, Australia   [12]Institute of Health and Biomedical Innovation, Queensland University of Technology, Translational Research Institute, Woolloongabba, Australia   [13]Department of Clinical Neuroscience, Karolinska Institute, Stockholm, Sweden   [14]Guy's and St Thomas' NHS Foundation Trust and King's College London NIHR Biomedical Research Centre, London, England

Correspondence: helmut.butzkueven@monash.edu
*Melissa M Gresle, Margaret A Jordan, Jim Stankovich, Judith Field, Alan G Baxter, and Helmut Butzkueven contributed equally to this work

**Table 1. Multiple Sclerosis case and healthy control demographics and sample numbers.**

| Measure | Case | Control |
|---|---|---|
| Total (n) | 73 | 97 |
| Female:male ratio | 2.8:1 | 1.9:1 |
| Median age (yr; range) | 39.2 (20–65) | 36.2 (21–64) |
| Median disease duration (yr; range) | 7 (0.1–36) | — |
| Median EDSS (range) | 2.0 (0–6.0) | — |
| Sample numbers for monocyte cells | 53 | 78 |
| Sample numbers for NK cells | 45 | 78 |
| Sample numbers for B cells | 37 | 87 |
| Sample numbers for CD4 cells | 38 | 85 |
| Sample numbers for CD8 cells | 55 | 91 |

acting at a distance referred to as trans-eQTL. As many MS risk SNPs are located near immune-associated genes, some of these could alter immune gene expression and contribute to immune heterogeneity (International Multiple Sclerosis Genetics Consortium, 2011). In a pivotal study by Raj et al (2014), purified monocyte and CD4-positive T cells from healthy individuals were used to conduct eQTL mapping of several candidate SNPs associated with the risk of neurodegenerative and autoimmune diseases, including MS. In their study, it was demonstrated that disease risk–associated cis-eQTLs were more commonly cell type specific compared with non-disease–associated cis-eQTLs and that MS risk–associated cis-eQTLs were overrepresented in CD4 T cells. The authors, therefore, suggested that genetic regulation in CD4 T cells could determine the immune contribution to MS risk.

Although eQTL mapping studies using cells from healthy individuals have greatly extended our knowledge of the likely functions of MS risk SNPs and their contributions to immune heterogeneity, there is now emerging evidence that regulatory variants can gain or lose their influence on transcriptional regulation in response to pro-inflammatory cytokine stimulation and other acquired cell states (Fairfax et al, 2012; Kim et al, 2014; Lee et al, 2014; De Jager et al, 2015). Hence, there is a clear need to consider the effects of disease risk variants both in the context of cell type and disease status. In the current study, we comprehensively investigated transcriptional changes associated with MS risk SNPs in recirculating leukocytes. We evaluated the effects on gene expression in purified immune cell types from both untreated MS cases and healthy controls.

# Results

## Patient demographics and sample acquisition

A total of 73 MS cases and 97 unaffected controls were included in the analysis, and their demographics are summarized in Table 1. Of the 73 cases, 3 were classed as clinically isolated syndrome (2/3 have since converted to relapsing-remitting multiple sclerosis), two progressive relapsing, and 64 relapsing-remitting multiple sclerosis. Note that no clinical data were available for two cases.

Five types of immune cells were investigated: monocytes, NK cells, B cells, CD4 cells, and CD8 cells. Not all cell types were analysed for all

individuals. The number of available datasets for each cell type was determined by blood volumes collected, individual cell yields, and cell purity (Table 1).

## Cis-eQTL associations for MS cases and unaffected controls

Each cell type was analysed separately. In our initial studies, we combined case and control datasets for each cell type to search for cis eQTL associations between 172 non-MHC MS risk SNPs (some correlated with each other) and 1,538 genes with transcription start sites within ±500 kb of a risk SNP. We analysed 2,711 SNP–gene pairs in each cell type, modeling log expression as a linear function of number of MS risk alleles carried (0, 1, or 2) and adjusting for case–control status. The associations with false discovery rate (FDR) q < 0.05 are listed in Supplemental Data 1 for each cell type. The numbers of genes per cell type varied from 33 in NK cells to 57 in CD8 T cells. In total, the list comprises 129 genes, including 53 genes whose expression is associated with MS risk genotype in more than one cell type (Fig 1). For all but 4 of these 53 genes, the directions of association were the same across cell types; for three of the four exceptions (MANBA, ADCY3, and JAZF1), the direction of association differed between monocytes and other cell types. Gene expression associations were identified for 100 of the 172 SNPs. We tested many pairs of SNPs in LD with one another; grouping SNPs in LD $r^2$ > 0.5 together leaves 128 groups of SNPs (Supplemental Data 2), of which 68 groups were eQTLs in at least one cell type (Table 2).

The MS risk SNPs were genotyped using Illumina's Immunochip custom array, which includes assays for many other SNPs near MS risk SNPs. For 42% (136/321) of the MS risk SNP–gene pairs identified with q < 0.05 across the five cell types, the MS risk SNP was the most significant eQTL on the Immunochip for that gene, or in strong linkage disequilibrium ($r^2$ > 0.8) with the most significant eQTL for the gene. In other cases, we adjusted associations between the MS risk SNP and the gene for genotypes at the most significant eQTL. In 28% of cases (89/321), the coefficient of the MS risk SNP was reduced by more than 70%, or changed sign. For the remaining 30% of cases (96/321), the coefficient of the MS risk SNP was reduced by less than 70% (Supplemental Data 3). Data for individual MS risk SNP–gene pairs are presented in Supplemental Data 4.

For each cell type, the top ranked association with an MS risk SNP is graphically represented in Fig 2A–E. Gene pathway analysis was performed based on lists of associated genes for each cell type, using Ingenuity Pathway Analysis software (QIAGEN Inc., https://www.qiagenbioinformatics.com/products/ingenuity-pathway-analysis). Genes in a few canonical pathways were enriched among the lists of top-ranked genes for each cell type, but the number of genes within these pathways was relatively small and, hence, provided limited functional insight (Supplemental Data 5).

## Effects of disease status on eQTLs

Among these cis eQTLs, there are a few with differences in expression between cases and controls. Expression of TUBD1 in NK cells was 7.7% higher in cases than controls after adjustment for genotype at rs180515 (P = 0.001, FDR q = 0.05 after adjusting for multiple testing of 45 SNP–gene pairs in NK cells) (Fig 3A). For two of the strongest eQTLs, there is evidence of genotype–phenotype interaction, where the eQTL effect differs between cases and controls. The rs703842

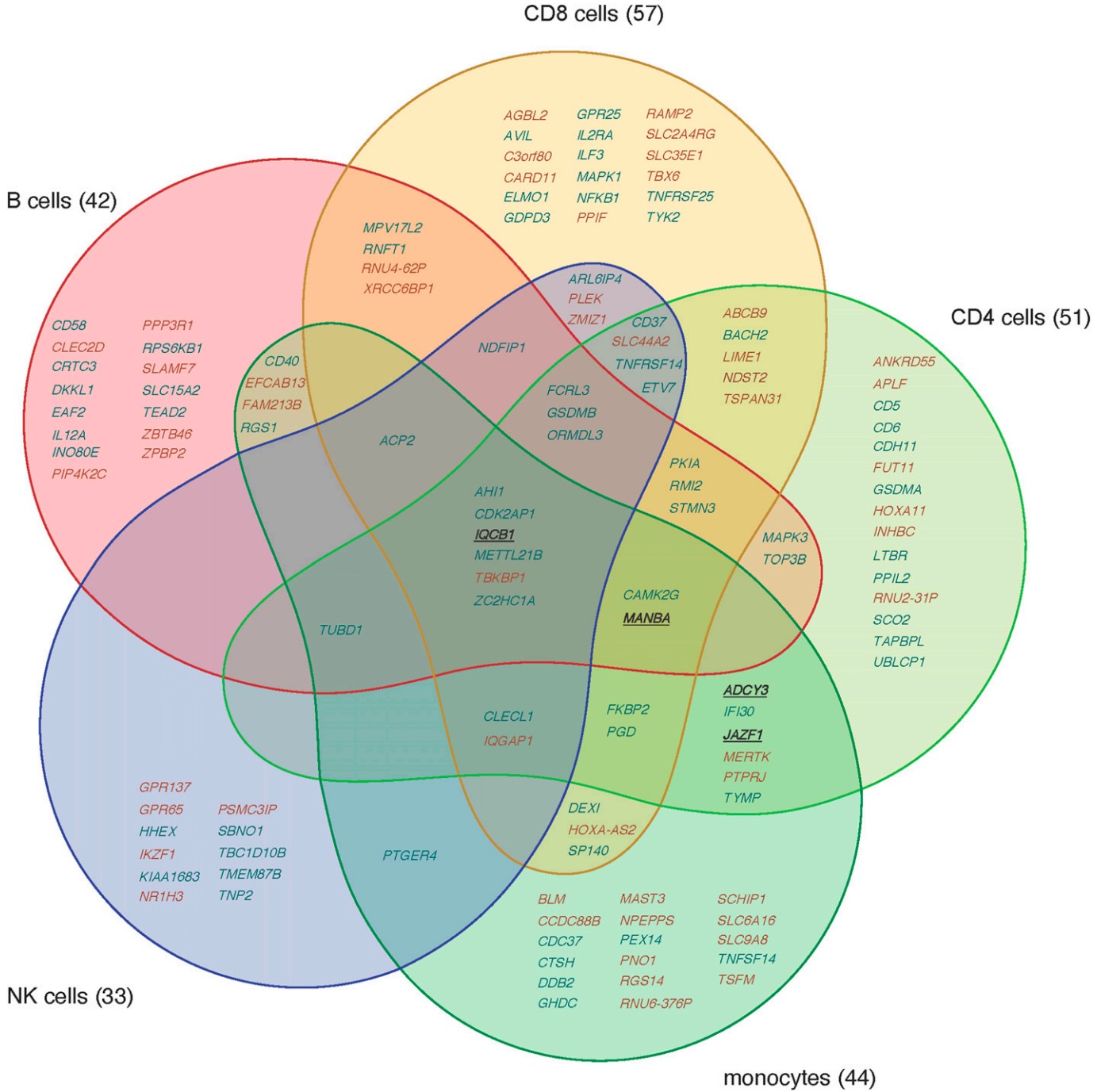

**Figure 1.  A graphical representation of expression quantitative trait loci genes by cell type.**
This Venn diagram depicts the genes (gene symbol given) associated with one or more multiple sclerosis (MS) risk single-nucleotide polymorphisms, per cell type. Here, we identify both cell type–specific and shared expression quantitative trait loci in each of the cell types assayed. Genes where the MS risk allele is associated with higher expression are colored brown, genes with the MS risk allele is associated with lower expression are colored blue, and genes where the direction of expression change varies between cell types are underlined.

MS risk allele (A) is associated with lower expression of the methyltransferase-like 21B (*METTL21B*) gene in CD8 cells of both cases and controls. However, the effect is stronger in cases (*P* = 0.0003 for genotype–phenotype interaction, FDR q = 0.03 after adjustment for multiple testing of 83 SNP–gene pairs in CD8 cells). In particular, cases

that are homozygous for the risk allele have lower expression of *METTL21B* relative to controls of the same genotype (Fig 2F). Similarly, the rs2760524 risk allele (G) is associated with lower expression of the regulator of G protein signaling 1 (*RGS1*) gene in monocytes of both MS cases and healthy controls; however, the effect of this allele is smaller

**Table 2. Summary of the number of reported expression quantitative trait loci associations by cell type.**

| Cell type | Total eQTL associations reported (SNP–gene pairs)[a] | No. of LD groups[b] | No. of individual genes[c] | No. of cell-specific LD groups[b] | No. of cell-specific genes[c] |
|---|---|---|---|---|---|
| Monocytes | 71 | 37 | 44 | 6 | 17 |
| NK cells | 45 | 27 | 33 | 1 | 11 |
| B cells | 54 | 27 | 42 | 1 | 15 |
| CD4 cells | 68 | 44 | 51 | 9 | 15 |
| CD8 cells | 83 | 41 | 57 | 4 | 18 |

[a]The SNP–gene pairs are listed in Supplemental Data 3.
[b]The LD groups are listed in Supplemental Data 2.
[c]The genes are listed in Supplemental Data 1 and Fig 1.

in MS cases ($P$ = 0.002 for genotype–phenotype interaction, FDR q = 0.1 after adjustment for multiple testing of 71 SNP–gene pairs in monocytes, Fig 2A). Table 3 shows all other SNP–gene pairs among the eQTLs with evidence of genotype–phenotype interaction (unadjusted $P <$ 0.05). Interestingly, two candidate SNPs that were not identified as eQTL in additive linear models using the combined case and control dataset were found to have significant genotype by phenotype interaction terms. The rs2256814 MS risk allele (A) appeared to have opposing effects on gene expression in CD4 cells of MS case relative to controls, associated with lower expression of the Myelin transcription factor 1 (*MYT1*) gene in MS cases (q = 0.05, adjusted for 2,711 pairs) (Fig 3B). Similarly, the rs12087340 MS risk allele (A) was associated with lower expression of the *RF00136* gene in monocytes of MS cases and higher expression in controls (q = 0.04) (Fig 3C).

### Comparisons of expression between cases and controls

In addition to analyses incorporating genotypes at MS risk SNPs, we also ran simple tests of differential expression between cases and controls. Among all genes within 500 kb of MS risk SNPs, expression of *SOCS1* in B cells was 16% higher in cases than controls ($P$ = 2 × $10^{-5}$, FDR q = 0.03 after adjustment for testing 1,538 genes near MS risk SNPs; Fig 3D). After correction for multiple testing, there were no other significant differences for genes near MS risk SNPs. Searching all transcripts represented on the microarray chip, expression of *SESN1* in B cells was 41% higher in cases than controls ($P$ = 4 × $10^{-8}$, FDR q = 0.001 after adjustment for testing 33,297 genes, Fig 3E) and expression of *FKBP5* in CD4 T cells was 22% higher in cases than controls ($P$ = 2 × $10^{-8}$, FDR q = 0.0007, Fig 3F). Full results of the case–control comparisons are shown in Supplemental Data 6.

## Discussion

Our comprehensive eQTL mapping studies reveal a significant contribution of MS susceptibility loci to genetically determined immune heterogeneity. Here, we describe eQTL associations for 129 genes in one or more immune cell types. For most of these genes, expression is influenced by multiple cis eQTLs, not just MS risk SNPs. Furthermore, for some of the associations we report, it is unlikely that MS susceptibility alleles influence gene expression directly, particularly when the association is substantially reduced after adjusting for genotypes at a nearby, stronger eQTL.

Interestingly, in contrast to previous reports by Raj et al (2014), we did not find evidence for a predominance of CD4 T-cell–specific MS risk eQTL associations within our studied population of MS cases and controls. Although Fig 2 probably overrepresents the number of cell-type–specific eQTLs because of the imposition of a fixed significance threshold, it nevertheless suggests that cell-type–specific eQTLs are relatively evenly distributed across innate and adaptive immune cell types. Based on these observations, we, therefore, propose that both the innate and adaptive arms of the immune system are likely to be important in determining individual susceptibility to MS.

Using multivariable linear regression models, we provide preliminary evidence that a small number of MS risk variants have differential effects on gene expression in MS cases compared with healthy controls. This is in line with recent observations by James et al (2018), who used monocytes isolated from healthy controls to show that some MS risk SNPs exert differential effects on gene expression after stimulation of these cells with interferon-γ or lipopolysaccharide. The context-specific nature of some eQTLs has also been demonstrated more widely both in vitro (Fairfax et al, 2014; Hu et al, 2014; Kim et al, 2014; Lee et al, 2014) and in vivo (Peters et al, 2016). Earlier eQTL mapping studies of disease risk loci were restricted to cells from healthy individuals (e.g., Raj et al [2014]) and, thus, could not assess if MS risk eQTL can be modified by disease-associated factors. Here, we describe at least one MS risk eQTL association, with a differential effect size in MS cases relative to controls. In CD8 cells, we show that rs703842 risk allele is associated with lower *METTL21B* expression in cases relative to controls. It has been reported that the encoded METTL21B protein functions as a methyltransferase, catalysing methylation on lysine 165 of the eukaryotic elongation factor 1A (eEF1A) (Hamey et al, 2017; Matecki et al, 2017). The eEF1A protein is a subunit of the eEF1A translation complex, which is responsible for the enzymatic delivery of aminoacyl tRNAs to the ribosome and is, thus, ubiquitously expressed in the context of protein synthesis. METTL21B is also reported to have roles in actin organization, apoptosis, the nuclear export of tRNAs, the guidance of damaged or misfolded proteins to the proteasome, viral propagation, microRNA biogenesis, and RNA interference (Reviewed in: Sasikumar et al [2012] and Yi et al [2015]). Hence, reduced expression of *METTL21B* gene expression in CD8 cells of MS cases could potentially affect a wide range of molecular pathways in these cells.

We also report two associations, rs2256814 *MYT1* in CD4 cells and rs12087340 *RF00136* in monocyte cells, with opposing effects on gene expression in MS cases compared with controls. Interestingly, these associations were not detectable where log expression was

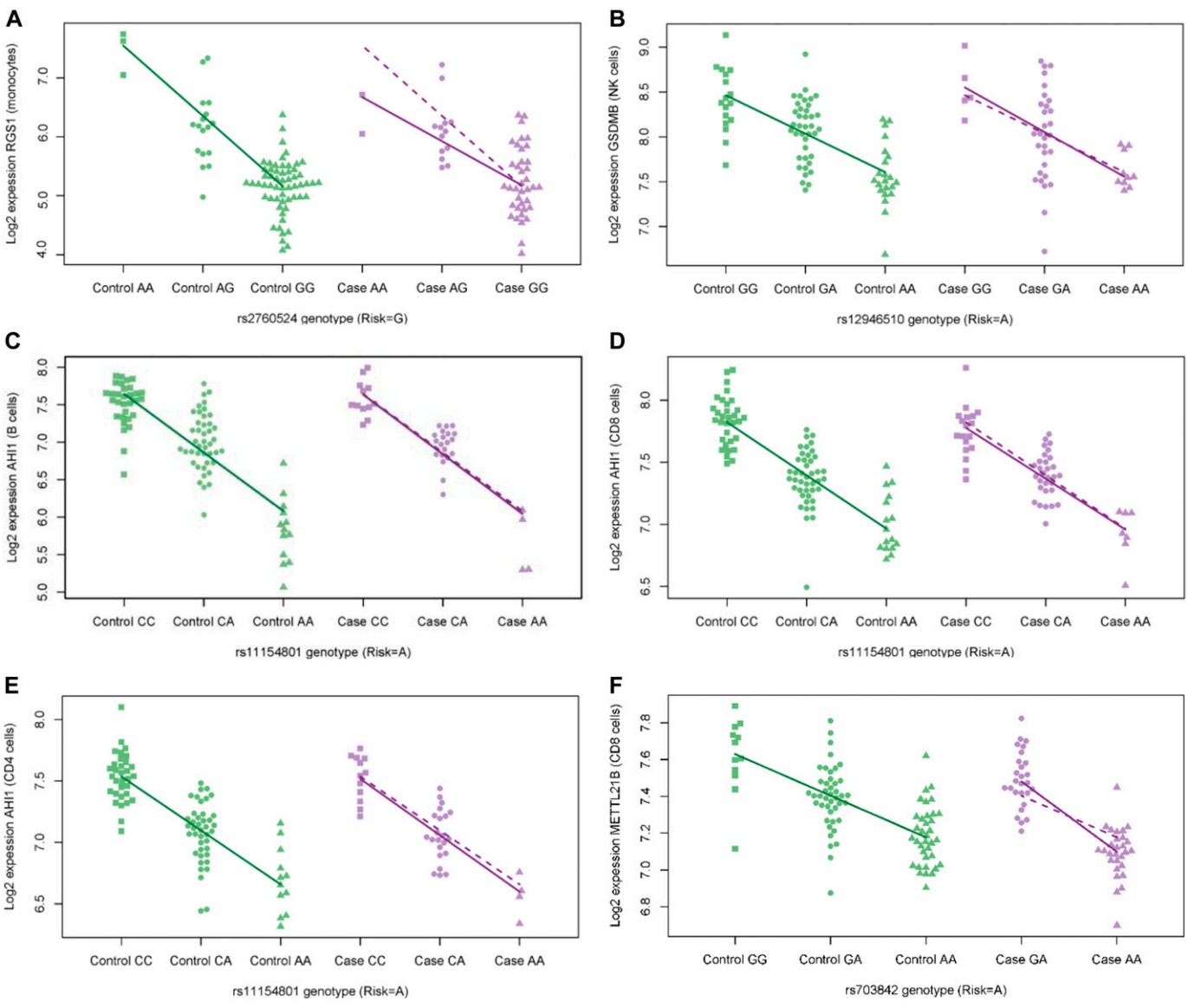

**Figure 2. The top ranked expression quantitative trait loci (eQTL) associations.**
Graphical representation of the top ranked eQTL associations at non-human leukocyte antigen multiple sclerosis (MS) risk loci, for each immune cell type, sorted by false discovery rate. For each eQTL association, log₂ gene transcript expression is shown for MS cases and healthy controls, segregated by single-nucleotide polymorphism (SNP) genotype. The regression lines in each figure demonstrate the associations between genotype and gene expression for each SNP/gene pair. Associations for controls (solid green lines) are superimposed on case plots (dashed purple lines) to facilitate comparison with case associations (solid purple lines). **(A)** In figure (A) the regulator of G-protein signaling 1 (*RGS1*) transcript log₂ expression versus rs2760524 genotype is shown for monocyte cells (risk genotype = G; q = 2 × 10⁻⁴⁷). Interestingly, there also appears to be a difference in the effect size of genotype on *RGS1* gene expression, between MS cases and controls: each additional copy of the G allele is associated with a 56% reduction of expression in controls, but only a 40% reduction of expression in cases (q = 0.1 for difference). **(B)** In figure (B), the Gasdermin B (*GSDMB*) log₂ transcript expression versus rs12946510 (risk genotype = A; q = 2 × 10⁻³⁶) genotype is shown for NK cells. **(C, D, E)** The Abelson helper integration site 1 (*AHI1*) log₂ transcript expression versus rs1115480 genotype 1 (risk genotype = A; q = 3 × 10⁻⁷²) is shown for MS cases and controls in B cells (C), CD8 cells (q = 3 × 10⁻⁷¹) (D), and CD4 cells (q = 6 × 10⁻⁷⁷) (E). **(F)** In (F) graphing the association between rs703842 genotype (risk genotype = A) and *METTL21B* log₂ transcript expression in CD8 cells, it reveals a difference in effect of the risk allele on gene transcript expression in MS cases relative to controls (genotype-by-phenotype interaction q = 0.03 adjusted for 83 eQTL SNP/gene pairs).

simply modeled as a linear function of the number of MS risk alleles carried in the combined case and control dataset, likely because of the variance produced by these differential regulatory effects. Little is known about the functions of the small nucleolar RNA encoded by *RF00136*; however, MYT1 protein is a known transcription factor that has been shown to bind the promoter region of the myelin proteolipid protein gene (Kim & Hudson, 1992). The MYT transcription factor has been reported to play a role in oligodendrocyte development and

possibly remyelination (Nielsen et al, 2004; Vana et al, 2007), but may also have wider functions in neural cell development (Yokoyama et al, 2014). The expression of MYT1 is widely reported as being restricted to adult neural cells; however, Vana et al (2007) did report MYT1 protein expression in some infiltrating lymphocytes in MS lesional tissue. Although the functions of the MYT1 protein in lymphocyte cells remain unknown, the observed association between the rs2256814 MS risk SNP genotype and *MYT1* gene expression in CD4 cells in our dataset raises

**Table 3. Expression quantitative trait loci with some evidence of genotype–phenotype interaction.**

| Cell type | SNP | Gene | Genotype q-value | Genotype × phenotype P-value |
|---|---|---|---|---|
| Monocytes | rs2760524 | RGS1 | $2 \times 10^{-47}$ | $2 \times 10^{-3}$ |
| Monocytes | rs1323292 | RGS1 | $2 \times 10^{-47}$ | $2 \times 10^{-3}$ |
| Monocytes | rs1359062 | RGS1 | $6 \times 10^{-45}$ | $1 \times 10^{-2}$ |
| Monocytes | rs533646 | RNU6-376P | $8 \times 10^{-5}$ | $3 \times 10^{-2}$ |
| Monocytes | rs4665719 | ADCY3 | $1 \times 10^{-2}$ | $5 \times 10^{-2}$ |
| Monocytes | rs11052877 | CLECL1 | $5 \times 10^{-4}$ | $5 \times 10^{-2}$ |
| NK cells | rs4648356 | TNFRSF14 | $1 \times 10^{-2}$ | $3 \times 10^{-2}$ |
| B cells | rs12946510 | ORMDL3 | $1 \times 10^{-20}$ | $7 \times 10^{-3}$ |
| B cells | rs12946510 | GSDMB | $4 \times 10^{-17}$ | $2 \times 10^{-2}$ |
| B cells | rs180515 | TUBD1 | $4 \times 10^{-3}$ | $3 \times 10^{-2}$ |
| CD4 cells | rs1021156 | PKIA | $1 \times 10^{-6}$ | $8 \times 10^{-3}$ |
| CD4 cells | rs2288904 | SLC44A2 | $1 \times 10^{-5}$ | $2 \times 10^{-2}$ |
| CD4 cells | rs703842 | METTL21B | $4 \times 10^{-22}$ | $3 \times 10^{-2}$ |
| CD4 cells | rs12212193 | BACH2 | $2 \times 10^{-3}$ | $4 \times 10^{-2}$ |
| CD4 cells | rs201202118 | METTL21B | $7 \times 10^{-27}$ | $4 \times 10^{-2}$ |
| CD8 cells | rs703842 | METTL21B | $3 \times 10^{-40}$ | $3 \times 10^{-4}$ |
| CD8 cells | rs201202118 | METTL21B | $1 \times 10^{-47}$ | $1 \times 10^{-3}$ |
| CD8 cells | rs9989735 | SP140 | $5 \times 10^{-2}$ | $8 \times 10^{-3}$ |
| CD8 cells | rs941816 | ETV7 | $2 \times 10^{-4}$ | $2 \times 10^{-2}$ |
| CD8 cells | rs949143 | ARL6IP4 | $3 \times 10^{-3}$ | $3 \times 10^{-2}$ |

SNP–gene pairs with an unadjusted genotype by phenotype P-value less than 0.05 are shown.

the intriguing possibility that this SNP could regulate *MYT1* gene expression in neural cells of MS cases in a similar way.

Interestingly, after adjustment for the SNP genotype, only one eQTL gene was found to be differentially expressed in MS cases relative to controls, *TUBD1* in NK cells. Hence, our findings suggest that these genes may not contribute to the disease process itself. Our transcriptome-wide analyses, without adjustment for genetic effects, identified only a few genes that were clearly differentially expressed in immune cells of the MS cases versus controls. Our data suggest that the functional state of immune cells isolated from peripheral blood, in this population of untreated MS cases with relatively early and stable disease, is similar to that of healthy controls. Two caveats are that potentially causative molecular variation is probably only present in a small subset of all cells of a specific population (e.g., CD4 T cells) and that these may not be detectable in the peripheral blood compartment. Our experimental design was enriched for eQTL associations detectable in the resting immune state in MS cases and healthy controls. Considering the emerging evidence for context-dependent activity at some genetic loci, additional detailed eQTL mapping studies using more refined immune cell subsets or using cells isolated during acute inflammatory relapse, or from lymph nodes, efferent lymph or CSF, could reveal regulation of gene expression in risk loci that were not captured using our experimental design. It is also possible that we may have missed some specific SNP transcriptional effects such as splicing and noncoding RNA using this microarray based approach.

Because of limited sample sizes and complexities associated with multiple testing, we acknowledge that our novel observations must be independently validated. However, we suggest that the overall signal of differential regulatory effects in MS cases compared with unaffected controls is likely to be robust, even if some of the individual signals may not be replicated. It also remains unclear how the functions of these risk variants are regulated in MS cases to produce these disease specific patterns of expression, if these disease associated differences in eQTL effect size are a cause or consequence of disease onset, or if they contribute to an abnormal immune response. Our results confirm, however, that there is a genetic contribution to immune heterogeneity in MS cases that is conveyed by MS risk SNPs and is detectable in the absence of overt immune activation. It is hoped that these studies will inspire a more concerted effort to capture genetically determined immune variation in the context of complex human disease, taking advantage of developing technologies to increase sensitivity and specificity.

# Materials and Methods

## Study population and approvals

Our study population included MS cases recruited from Box Hill Hospital, Victoria, between 2009 and 2013. Diagnoses were made by MS neurologists according to the McDonald (2001) criteria. Cases were not relapsing at the time of sample collection and were not treated with disease-modifying drugs or immunosuppressants for at

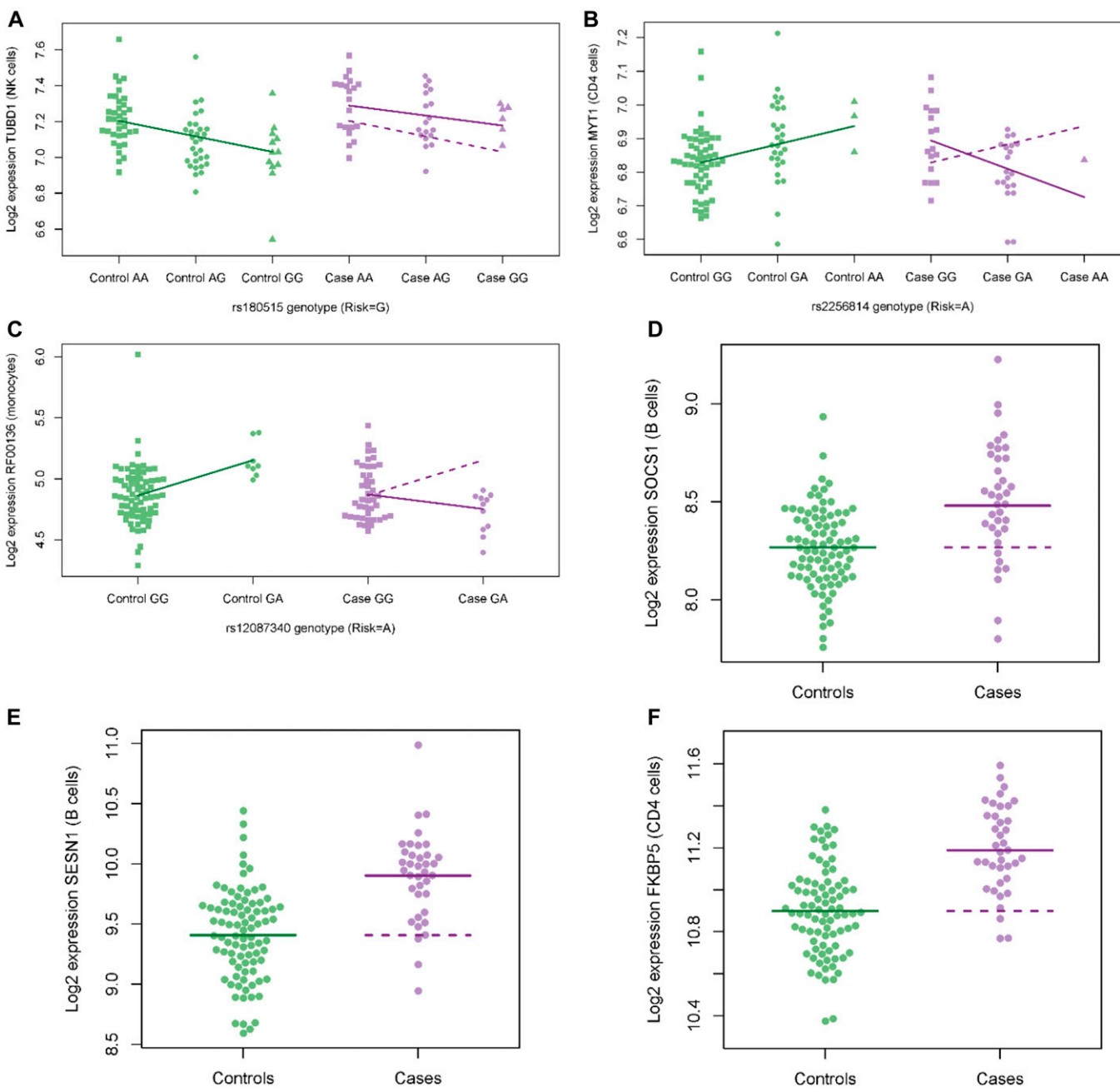

**Figure 3. Multiple sclerosis (MS) case and control differences in gene expression.**
**(A)** In (A) the expression of tubulin delta 1 (*TUBD1*) in NK cells is higher in MS cases relative to controls after adjustment for rs180515 genotype (risk allele G; q = 0.05 adjusting for 45 expression quantitative trait loci associations). **(B)** The rs2256814 risk allele A is associated with lower expression of the myelin transcription factor 1 (*MYT1*) gene in MS cases and higher expression in controls (genotype-by-phenotype interaction q = 0.05 adjusted for 2,711 pairs). **(C)** Similarly, the rs12087340 risk allele A is associated with lower expression of the *RF00136* gene in MS cases and higher expression in controls (q = 0.04 adjusted for 2,711 pairs). **(D, E, F)** The *SOCS1* gene in B cells, localized to the middle of a cluster of MS risk single-nucleotide polymorphisms on chromosome 16, (E) *SESN1* in B cells, and (F) *FKBP5* in CD4 T cells. In (A, B, C), the regression lines for controls (solid green lines) are superimposed on case plots (dashed purple lines) to facilitate comparison with case associations (solid purple lines). In (D, E, F), the mean $log_2$ expression values for controls (solid green lines) are superimposed on case plots (dashed purple lines) to facilitate comparison with case means (solid purple lines).

least 1 mo before participating in the study. Unaffected controls were generally allied health staff, nurses, research scientists, and spouses/partners of MS patients, who had no known neurological or autoimmune diseases. The age and gender of unaffected controls were also recorded. All MS and healthy control participants were selected for European ancestry, to minimize confounding by ancestry and because most Australian patients with MS are of European ancestry. The case and control demographics are summarized in Table 1.

The study was conducted according to the Declaration of Helsinki principles and was approved by the Human Research Ethics

Committees from the participating hospital. Written informed consent was obtained from all participants.

## Isolation of circulating peripheral immune cells and RNA preparation

A venous blood sample of up to 110 ml was collected into EDTA-coated blood collection tubes (Becton Dickinson), from each participant between 7:30 AM and 11:00 AM, to minimize the influence of circadian variation. For isolation of PBMCs, B cells, and CD4 and CD8 T cells, up to 70 ml of blood was diluted twofold with PBS (dPBS, 2 mM EDTA, and 2% FCS) and slowly layered on to 15 ml Histopaque-1077 in 50-ml tubes (Sigma-Aldrich). The samples were centrifuged at 400$g$ (brakes off) for 30 min at room temperature and then the PBMC layer was removed using a glass pipette. The cells were washed twice with PBS solution, counted, and incubated with human CD19, and CD8 microbeads for on-column (LS) positive selection of B cells and CD8 T cells, respectively, according to the manufacturers' instructions (Miltenyi Biotech). The negative fractions from the CD8 cell separation were used to isolate CD4 T cells by magnetic cell sorting using CD4 microbeads (Miltenyi Biotech), according to the manufacturer's instructions. For isolation of monocytes and NK cells, 40 ml of blood was centrifuged at 400$g$ (brakes off) for 15 min at room temperature, and the buffy coat layer was removed using a glass pipette. The buffy coat was then divided into two portions for incubation with RosetteSep human monocyte and NK cell enrichment cocktails (Stemcell Technologies), respectively. Each portion was then layered on to 5 ml Histopaque in 15-ml tubes, centrifuged at 1,200$g$ (brakes off), and the enriched cell layers were removed using a glass pipette. The cells were washed, counted, and labelled with human CD14 (monocytes) or CD56 (NK cells) microbeads for positive cell selection using an autoMACS Pro separator (Miltenyi Biotech). After isolation, purity was checked using standard flow cytometry protocols on a Cyan Flow cytometer (Beckman Coulter), and only samples that were >90% pure were included. We isolated immune cell type RNA using RNeasy mini kits (QIAGEN), and the RNA yield was quantified spectrophotometrically on a NanoDropND-1000.

## Microarray processing

Using 100 ng of RNA/sample, expression microarray hybridizations were performed using the WT Expression kit (Life Technologies), WT Terminal Labelling and Controls Kit (Affymetrix), and Affymetrix Human Gene_1.0ST arrays, which contain 764,885 distinct probes. The probed arrays were washed and stained using the GeneChip Hybridization Wash and Stain Kit (Affymetrix) and scanned using the GeneChip Scanner 3000. Images (.dat files) were processed using GeneChip Command Console (Affymetrix) and CEL files imported into Partek Genomics Suite 6.6 (Partek SG) for further quality control checks.

## Genotyping

Whole blood genomic DNA was isolated using the Illustra Nucleon BACC3 kit according to the manufacturer's instructions. All MS cases and unaffected controls were genotyped using the Immunochip custom array (Illumina) in accordance with Illumina protocols, at the University of Queensland Diamantina Institute (Brisbane, Australia). Genotype calling

was performed using lllumina iScan System and the Genotyping Module (v.1.8.4) of the GenomeStudio Data Analysis software.

## Genotyping quality control

PLINK was used to perform linkage disequilibrium-based SNP pruning and to estimate pairwise identity-by-descent between individuals using the pruned SNP set (Purcell et al, 2007). Four close relative pairs were identified (second-degree relatives or closer), and one individual from each relative pair was removed from the dataset. In a principal components analysis of the pruned genotypes, there were no outlier samples more than six SDs from the mean along the first 10 principal components. Testing for differences in allele frequencies between MS cases and controls, the estimated genomic inflation factor ($\lambda$) was 0.9986.

## Selection of MS risk SNPs to test for association with gene expression

A total of 183 SNPs outside the MHC region, previously identified in large-scale genome wide association studies as significantly or strongly suggestive of being associated with MS risk (International Multiple Sclerosis Genetics Consortium et al 2011, 2013; Patsopoulos et al, 2011), were considered for testing as candidate eQTLs (see Supplemental Data 2 for a list of these SNPs). 16 of the 183 SNPs were not included in eQTL analyses: 12 because they were not on the Immunochip and 4 because genotypes were called in less than 95% of samples. For these 16 SNPs, the LDlink Web site was used to search for proxies on the Immunochip using 1000 Genomes Data from European populations (Machiela & Chanock, 2015). For 3 of the 16 excluded SNPs, the best proxy was an SNP already included in the list of MS risk SNPs. For another five SNPs, additional proxies were identified ($r^2$ between 0.59 and 1.00) and included in the analysis. For the remaining eight excluded SNPs, no proxy ($r^2 > 0.5$) was found on the Immunochip. Hence, analyses included 172 SNPs: 167 of the original 183 MS risk SNPs plus five additional proxy SNPs. Genotype call rates were greater than 98%, and Hardy–Weinberg equilibrium $P$-values were greater than 0.002 for all 172 SNPs. These SNPs cluster into 128 LD groups, with $r^2 > 0.5$ between SNPs in the same LD group (Supplemental Data 2).

## Processing and analysis of expression data

Expression datasets for each cell type were analysed separately throughout.

Expression measurements for samples of each cell type were preprocessed using the Robust Multi-array Average algorithm as implemented in the R package "oligo" (Carvalho & Irizarry, 2010), to perform background correction, quantile normalization, and to summarize expression at the "core gene" level (33,297 genes) for each sample.

The R package "Remove Unwanted Variation" (RUV) was used to test for associations between genotype, phenotype (MS case or control), and expression (Gagnon-Bartsch & Speed, 2012; Gagnon-Bartsch et al, 2013). Methods from this package use negative control genes to estimate factors of unwanted variation $W$. Then log gene

expression values $Y$ are modeled as functions of $W$ and factors of interest $X$:

$$Y = X\beta + W\alpha + \varepsilon.$$

A set of 575 housekeeping genes, mapping to 593 Affymetrix cluster IDs, was used as negative controls (Eisenberg & Levanon, 2003). RUV includes several methods to estimate factors of unwanted variation $W$ from negative controls, including methods called RUV-2, RUV-4, RUV-inv, and RUV-rinv. Performances of these methods were compared by applying them to our monocyte dataset and measuring the rates of detection of 193 "high confidence eQTLs" identified in two previous monocyte eQTL studies with associated SNPs on the Immunochip (Zeller et al, 2010; Fairfax et al, 2012; Yu et al, 2016). In the end, we settled on the ridged inverse method (RUV-rinv) with empirical variance estimates for all differential expression analyses. There were several reasons for choosing this method: diagnostic plots, the method's ability to detect high confidence monocyte eQTLs, its insensitivity to choice of negative controls (compared with RUV-2), and the advantage that it does not require estimation of the number of unwanted factors (unlike RUV-2 and RUV-4). For each of the five cell types and each of the 172 MS-associated SNPs, two RUV-rinv models were run: one with $X$ comprising two factors of interest (number of MS risk alleles $G$ and phenotype $P$) and one with $X$ comprising three factors ($G$, $P$, and genotype-by-phenotype interaction $G \times P$).

In addition to these eQTL analyses, a simple case–control comparison was run using RUV-rinv for each cell type, that is, with phenotype the only factor of interest ($X = P$).

### Significance thresholds to allow for multiple testing

To search for cis eQTLs, we examined coefficients of genotype $G$ in the two-factor models for genes whose transcription start sites lie within 500 kilobases of the 172 MS-associated SNPs (hg38 genome build). The FDR method of Benjamini and Hochberg was used to adjust for multiple testing of the 2,711 SNP–gene pairs that were identified. The same adjustment was applied to search the 2,711 SNP–gene pairs for associations with phenotype in the two-factor models, and for genotype–phenotype interactions in the three-factor models. We also looked for phenotype and interaction effects just among the cis eQTLs, applying a less stringent adjustment controlling for the number of eQTLs identified in each cell type.

For each gene whose expression was associated with genotypes at an MS risk SNP with FDR $q < 0.05$, we searched for other eQTLs within 500 kb of the transcription site of the gene. For each SNP on the Immunochip within this distance, the same RUV-rinv models were run as for the MS risk SNP. The SNP showing the most significant association with the gene was recorded and linkage disequilibrium calculated between this SNP and the MS risk SNP. For genes where $r^2$ between genotype $G_{sig}$ at the most significant SNP and genotype $G_{MS}$ at the MS risk SNP was less than 0.8, we ran an RUV-rinv model with genotypes at both SNPs included:

$$Y = G_{MS}\beta_{MS-adj} + G_{sig}\beta_{sig} + P\beta_P + W\alpha + \varepsilon.$$

This calculates the effect $\beta_{MS-adj}$ of the MS risk SNP on log gene expression adjusted for genotypes at the most significant SNP. This coefficient was compared with the unadjusted coefficient $\beta_{MS-unadj}$ estimated from the original RUV-rinv model:

$$Y = G_{MS}\beta_{MS-unadj} + P\beta_P + W\alpha + \varepsilon.$$

To search for genes differentially expressed between MS cases and controls without consideration of genotypes, we searched both among the 1,538 genes within 500 kb of the 172 MS-associated SNPs and transcriptome wide, applying appropriate FDR corrections in the two cases.

The R package "venn" (Dusa, 2018) was used to plot a Venn diagram showing which genes' expression levels were associated with MS risk SNPs in which cell types.

### Cross-normalization

Expression levels were plotted for selected SNP–gene pairs to show genotype, phenotype, and genotype-by-phenotype effects after removal of unwanted variation. Estimating and subtracting the term $W\alpha$ for unwanted variation results in over-adjustment (data not shown). A simple variant of this strategy, called cross-normalization, is very effective (Y Pan and J Gagnon-Bartsch, manuscript in preparation). As in cross-validation, cross-normalization omits each observation (here case or control sample) in turn and estimates the quantities needed for removing the unwanted variation from that omitted observation, using all the other observations. Cross-normalized expression values were calculated by applying RUV-rinv with design matrix $X = [G|P|G \times P]$ or $X = P$ and were plotted using the R package "beeswarm" (Eklund, 2016). Cross-normalization with $X = P$ was used to calculate adjusted expression values for making relative log expression and singular value decomposition plots. Comparison of these plots before and after cross-normalization demonstrated the presence and removal of batch effects in the expression data (Supplemental Data 7).

### Pathway analyses

Gene ontology and biological pathway analyses were conducted using Ingenuity Pathway Analysis Software (QIAGEN Inc., https://www.qiagenbioinformatics.com/products/ingenuity-pathway-analysis) using lists of genes whose expression was significantly associated with MS risk SNPs in each cell. These analyses were not performed for transcriptome-wide case versus control comparisons, as relatively few genes were within the FDR cutoff of <0.05.

## Data Availability

The CEL files, Robust Multi-array Average-adjusted gene expression values for all tested transcripts, and SNP genotype data generated in this study have been deposited at the European Genome-phenome Archive (EGA), which is hosted by the European Bioinformatics Institute and the Centre for Genomic Regulation, under accession number EGAS00001004087.

## Supplementary Information

# Acknowledgements

We thank all the people with MS and control persons who participated in this study. We also thank Lisa Taylor, Sandra Williams, and K-J Lazarus for their assistance with the collection of blood samples for this study. We would like to acknowledge the support and encouragement of this work provided by Mr Anton and Mrs Dulcie Christensen of Sarina, Queensland, Australia. Thank you to Johann Gagnon-Bartsch and Yujia Pan for providing details of the cross-normalization method. This work was supported by NHMRC Project Grants (#1032486) and project grants from the Australian Research Council (LP110100473) Multiple Sclerosis Research Australia (#15-025), the Lions Clubs of Australia (titled "New models of Multiple Sclerosis Susceptibility" and "Multiple Sclerosis Research Program"), and an equipment grant from the Rebecca L Cooper Foundation (#10027). MM Gresle was supported by fellowships from Multiple Sclerosis Research Australia (#14-069) and the Melbourne Brain Centre (National Health and Medical Research Council [NMHRC] center for Research Excellence grant 1001216). MA Jordan was supported by an NHMRC/MSRA Betty Cuthbert fellowship. M Bahlo was supported by an NHMRC Program Grant (GNT1054618) and an NHMRC Senior Research Fellowship (GNT1102971). This work was supported by the Victorian Government's Operational Infrastructure Support Program and the NHMRC Independent Research Institute Infrastructure Support Scheme (IRIISS). MA Brown is supported by an NHMRC Senior Principal Research Fellowship (#1024879). AG Baxter was supported by an NHMRC Fellowship (#1003118). J Field was supported by a Multiple Sclerosis Research Australia Fellowship. H Butzkueven is supported by an NHMRC practitioner fellowship (#1080518).

## Author Contributions

MM Gresle: conceptualization, formal analysis, supervision, funding acquisition, investigation, methodology, project administration, and writing—original draft, review, and editing.

MA Jordan: conceptualization, formal analysis, supervision, funding acquisition, investigation, methodology, project administration, and writing—original draft, review, and editing.

J Stankovich: conceptualization, formal analysis, validation, methodology, and writing—original draft, review, and editing.

T Spelman: formal analysis, validation, methodology, and writing—review and editing.

LJ Johnson: formal analysis, investigation, and writing—review and editing.

L Laverick: formal analysis, investigation, and writing—review and editing.

A Hamlett: investigation and writing—review and editing.

LD Smith: investigation and writing—review and editing.

VG Jokubaitis: formal analysis and writing—review and editing.

J Baker: investigation and writing—review and editing.

J Haartsen: investigation and writing—review and editing.

B Taylor: conceptualization, funding acquisition, and writing—review and editing.

J Charlesworth: conceptualization, funding acquisition, methodology, and writing—review and editing.

M Bahlo: methodology and writing—review and editing.

TP Speed: formal analysis, methodology, and writing—review and editing.

MA Brown: formal analysis, investigation, methodology, and writing—review and editing.

J Field: conceptualization, formal analysis, supervision, funding acquisition, investigation, methodology, project administration, and writing—original draft, review, and editing.

AG Baxter: conceptualization, supervision, funding acquisition, methodology, project administration, and writing—original draft, review, and editing.

H Butzkueven: conceptualization, supervision, funding acquisition, methodology, project administration, and writing—original draft, review, and editing.

## Conflict of Interest Statement

H Butzkueven's institution received compensation for service on scientific advisory boards and as a consultant for Roche, Biogen, Merck, and Novartis and speaker honoraria from Biogen, Merck, Novartis, Roche, NHMRC, Medical Research Future Fund, and the Pennycook Foundation. MA Brown has received compensation for service on scientific advisory board and as a consultant for Abbvie, Ipsen, Janssen, Novartis, UCB, and Pfizer. MM Gresle has received honoraria from Biogen, Sanofi Genzyme, and Roche Australia. J Haartsen has received speaking and travel honoraria from Biogen Australia, Novartis Australia, Merck Australia, Roche Australia, and Genzyme Australia, and consultancy fees for nursing Advisory Boards for Biogen Australia, Merck Australia, Roche Australia, and Genzyme Australia. T Spelman has received compensation for serving on advisory boards and steering committees from Biogen.

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
