## [Reviewer comments · Life Science Alliance]

Life Science Alliance

Multiple Sclerosis risk variants regulate gene expression in innate and adaptive immune cells

Melissa Gresle, Margaret Jordan, Jim Stankovich, Tim Spelman, Laura Johnson, Louise Laverick, Alison Hamlett, Letitia Smith, Vilija Jokubaitis, Josephine Baker, Jodi Haartsen, Bruce Taylor, Jac Charlesworth, Melanie Bahlo, Terence Speed, Matthew Brown, Judith Field, Alan Baxter, and Helmut Butzkueven

DOI: <https://doi.org/10.26508/lsa.202000650>

Corresponding author(s): Helmut Butzkueven, Monash University, Central Clinical School, Department of Neuroscience

Review Timeline:	Submission Date:	2020-01-20
	Editorial Decision:	2020-03-16
	Revision Received:	2020-05-26
	Editorial Decision:	2020-05-26
	Revision Received:	2020-05-27
	Accepted:	2020-05-28

Transaction Report:

March 16, 2020

Re: Life Science Alliance manuscript #LSA-2020-00650-T

Prof. Helmut Butzkueven
Monash University, Central Clinical School, Department of Neuroscience
Level 6 Alfred Centre
99 Commercial Road
Melbourne, VIC 3004
Australia

Dear Dr. Butzkueven,

Thank you for submitting your manuscript entitled "Multiple Sclerosis risk variants regulate gene expression in innate and adaptive immune cells" to Life Science Alliance. The manuscript was assessed by expert reviewers, whose comments are appended to this letter.

As you will see, while the reviewers appreciate your data, some further analyses are needed to provide a significant value to others. We would thus like to invite you to submit a revised version of your manuscript to us, addressing the reviewer concerns. They provide constructive input and following their suggestions seems rather straightforward. But please do get in touch in case you would like to discuss individual revision points further.

Thank you for this interesting contribution to Life Science Alliance. We are looking forward to receiving your revised manuscript.

Sincerely,

Andrea Leibfried, PhD
Executive Editor
Life Science Alliance
Meyershofstr. 1
69117 Heidelberg, Germany
t +49 6221 8891 502
e a.leibfried@life-science-alliance.org
www.life-science-alliance.org

B. MANUSCRIPT ORGANIZATION AND FORMATTING:

Reviewer #1 (Comments to the Authors (Required)):

This interesting manuscript entitled "Multiple Sclerosis risk variants regulate gene expression in innate and adaptive immune cells." shows anew insight into MS risk loci that function by regulating gene expression in cell types relevant to MS.

In this study the authors describe eQTL associations for 129 genes in one or more immune cell types. Interestingly, in contrast to previous reports by Raj et al (2014), they did not find evidence for a predominance of CD4+ T-cell specific MS risk eQTL associations within the studied population of MS cases and controls. While Figure 2 probably over-represents the number of cell-type specific eQTLs due to the imposition of a fixed significance threshold, it nevertheless suggests that cell-type specific eQTLs are relatively evenly distributed across innate and adaptive immune cell types.

1] Patient demographics and sample acquisition

Question: Is there any specific to pick samples which have European ancestry ?
2] Cis-expression quantitative trait locus associations for MS cases and unaffected controls: Strongly supportive

3] Genotyping Quality Control: Strongly supportive

4] Processing and analysis of expression data: Strongly supportive

5] Pathway Analyses: Strongly supportive

I strongly recommend the editors to publish this manuscript.

Reviewer #2 (Comments to the Authors (Required)):

This is an interesting work, in which authors have mapped, at regions previously reported associated with MS, cis eQTL in monocytes, CD4, CD8, B-lymphocyte and NK cells from MS patients and healthy controls. They have found some MS associated variants that were eQTLs and some of them were differentially expressed between healthy and MS samples.

Authors search for cis eQTL associations between 172 non-MHC MS risk SNPs and the genes within +/- 500kb of a risk SNP.

Minor point: Author should clarify how many MS associated loci are analyzed. The text is confusing and gives the impression that the 172 are independent signals.

The authors focus only on MS risk variants to calculate the eQTLs and they miss a lot of interesting information. Since they use the immunochip to genotype the MS association regions, with very high density of markers per region, it would be an important opportunity to find new MS specific eQTLs. On the other hand, to determine if the MS associations and the eQTLs have the same origin, it is necessary to define the best eQTL for each gene in each locus.

Major point: Authors should calculate the correlation of all variants of the immunochip, in each specific locus and cell type, with the expression of the 129 genes that showed eQTLs with the MS risk variants. The authors should discuss if the MS risk variant and best eQTL for each gene come or not from the same origin.

Minor point. The author in the abstract refers to the variants that correlate with expression changes as "regulatory variants". Most of the eQTLs are not "regulatory", they are in LD with the causal variants. Authors do not show functional studies that could demonstrate that the analyzed SNPs were regulatory. This expression should be corrected through the paper.

Reviewer #1:

We thank Reviewer 1 for their time and positive comments. Our response to their question is given below:

Reviewer question 1: Is there any specific reason to pick samples which have European ancestry?

Author Response: Apologies that this was not clear. We focused on recruiting participants of European ancestry for this study to minimize confounding by ancestry, and because most Australian patients with MS are of European ancestry. This has been added to the manuscript (page 18).

Reviewer #2

We thank reviewer 2 for their constructive comments and suggestions. We provide a detailed response to each of these below:

Reviewer minor point 1: Author should clarify how many MS associated loci are analyzed. The text is confusing and gives the impression that the 172 are independent signals.

Author Response: Apologies that this was not clear. 172 SNPs were assessed, however a number of these were in LD with one or more SNPs, and so our analyses included 128 LD groups, with $r^2 > 0.5$ between SNPs in the same LD group. This information has been added to the description in the methods (page 21) to complement the information provided in supplementary data S1. We have also mentioned that some of the 172 SNPs are correlated with each other in the Results section (page 6).

Reviewer minor point 2: The authors focus only on MS risk variants to calculate the eQTLs and they miss a lot of interesting information. Since they use the immunochip to genotype the MS association regions, with very high density of markers per region, it would be an important opportunity to find new MS specific eQTLs. On the other hand, to determine if the

MS associations and the eQTLs have the same origin, it is necessary to define the best eQTL for each gene in each locus.

Author Response: Please see response to major point below.

Reviewer major point: Authors should calculate the correlation of all variants of the immunochip, in each specific locus and cell type, with the expression of the 129 genes that showed eQTLs with the MS risk variants. The authors should discuss if the MS risk variant and best eQTL for each gene come or not from the same origin.

Author Response: For each of the 129 genes that showed eQTLs with MS risk variants, we have now searched for other eQTLs within 500 kilobases of the transcription site of the gene. For each gene we have reported the best eQTL we found (the SNP with the smallest p-value) in Supplemental Data S2. When r^2 between the best eQTL and the MS risk variant is less than 0.8, we have calculated the reduction in the estimated effect of the MS risk SNP after adjusting for the best eQTL, and summarized our findings in a new figure (Supplemental Data S6). We have added a paragraph of text to the methods (page 23), and a paragraph of text to the results (page 7). We have also added two sentences to the first paragraph of the Discussion (page 15). We did not find evidence for new MS specific eQTLs at any of the new SNPs tested (no significant phenotype-genotype interaction terms, after correction for multiple testing).

Reviewer minor point 3. The author in the abstract refers to the variants that correlate with expression changes as "regulatory variants". Most of the eQTLs are not "regulatory", they are in LD with the causal variants. Authors do not show functional studies that could demonstrate that the analyzed SNPs were regulatory. This expression should be corrected through the paper.

Author Response: Agreed. We have modified this language throughout.

Please find the described changes to the manuscript file as tracked. Please also note that the abstract word count was reduced, figures were removed from the manuscript file and uploaded individually, and tables and figure legends were arranged to the end of the manuscript file as per your website instructions.

May 26, 2020

RE: Life Science Alliance Manuscript #LSA-2020-00650-TR

Prof. Helmut Butzkueven
Monash University, Central Clinical School, Department of Neuroscience
Level 6 Alfred Centre
99 Commercial Road
Melbourne, VIC 3004
Australia

Dear Dr. Butzkueven,

Thank you for submitting your revised manuscript entitled "Multiple Sclerosis risk variants regulate gene expression in innate and adaptive immune cells". I appreciate the introduced changes, and we would thus be happy to publish your paper in Life Science Alliance.

Please login one more time to fill in the electronic license to publish form. You will see that I changed the format of your supplementary files (please move all files to the next manuscript version) to "Supplementary Data Set 1-7" because in-line display of the files in the HTML version is too difficult. The files will thus be downloadable from the online version of the paper.

A. FINAL FILES:

-- Summary blurb (enter in submission system): A short text summarizing in a single sentence the study (max. 200 characters including spaces). This text is used in conjunction with the titles of papers, hence should be informative and complementary to the title. It should describe the context

and significance of the findings for a general readership; it should be written in the present tense and refer to the work in the third person. Author names should not be mentioned.

B. MANUSCRIPT ORGANIZATION AND FORMATTING:

Sincerely,

Andrea Leibfried, PhD
Executive Editor
Life Science Alliance
Meyerohofstr. 1
69117 Heidelberg, Germany
t +49 6221 8891 414
e contact@life-science-alliance.org
www.life-science-alliance.org

May 28, 2020

RE: Life Science Alliance Manuscript #LSA-2020-00650-TRR

Prof. Helmut Butzkueven
Monash University, Central Clinical School, Department of Neuroscience
Level 6 Alfred Centre
99 Commercial Road
Melbourne, VIC 3004
Australia

Dear Dr. Butzkueven,

Thank you for submitting your Resource entitled "Multiple Sclerosis risk variants regulate gene expression in innate and adaptive immune cells". It is a pleasure to let you know that your manuscript is now accepted for publication in Life Science Alliance. Congratulations on this interesting work.

DISTRIBUTION OF MATERIALS:

Again, congratulations on a very nice paper. I hope you found the review process to be constructive and are pleased with how the manuscript was handled editorially. We look forward to future exciting submissions from your lab.

Sincerely,

Reilly Lorenz
Editorial Office Life Science Alliance
Meyerhofstr. 1
69117 Heidelberg, Germany
t +49 6221 8891 414
e contact@life-science-alliance.org
www.life-science-alliance.org